# Role of NF-κB Signaling in the Interplay between Multiple Myeloma and Mesenchymal Stromal Cells

**DOI:** 10.3390/ijms24031823

**Published:** 2023-01-17

**Authors:** Marco Cippitelli, Helena Stabile, Andrea Kosta, Sara Petillo, Lorenzo Lucantonio, Angela Gismondi, Angela Santoni, Cinzia Fionda

**Affiliations:** 1Department of Molecular Medicine, Sapienza University of Rome, 00161 Rome, Italy; 2Istituto Pasteur-Fondazione Cenci Bolognetti, Sapienza University of Rome, 00161 Rome, Italy; 3IRCCS Neuromed, 86077 Pozzilli, Italy

**Keywords:** NF-kappa B, multiple myeloma, mesenchymal stromal cells

## Abstract

Nuclear factor-κB (NF-κB) transcription factors play a key role in the pathogenesis of multiple myeloma (MM). The survival, proliferation and chemoresistance of malignant plasma cells largely rely on the activation of canonical and noncanonical NF-κB pathways. They are triggered by cancer-associated mutations or by the autocrine and paracrine production of cytokines and growth factors as well as direct interaction with cellular and noncellular components of bone marrow microenvironment (BM). In this context, NF-κB also significantly affects the activity of noncancerous cells, including mesenchymal stromal cells (MSCs), which have a critical role in disease progression. Indeed, NF-κB transcription factors are involved in inflammatory signaling that alters the functional properties of these cells to support cancer evolution. Moreover, they act as regulators and/or effectors of pathways involved in the interplay between MSCs and MM cells. The aim of this review is to analyze the role of NF-κB in this hematologic cancer, focusing on NF-κB-dependent mechanisms in tumor cells, MSCs and myeloma–mesenchymal stromal cell crosstalk.

## 1. Introduction

The NF-κB family of transcription factors is a master regulator of many physiological and pathological processes, such as the immune response and tumorigenesis [1,2]. It comprises five proteins, NF-κB1 (p50), NF-κB2 (p52), RelA (p65), RelB and c-Rel, which are able to homo- or heterodimerize and bind a specific consensus DNA sequence on the promoter region of many target genes. The function of these transcription factors is regulated at different levels, including the cellular expression, the nuclear translocation and the transcriptional activity. They can be present in the cell as inactive precursors (p105 for p50 and p100 for p52) and their nuclear translocation is prevented by the interaction with a group of cytoplasmic inhibitory proteins belonging to the IκB family (IκBα, IκBβ, IκBγ, IκBε, IκBζ). A further level of complexity of NF-κB regulation is given by the existence of two main activating pathways: canonical (or classical) and noncanonical (or alternative). The former is triggered by activation of the kinase complex IKK (containing IKK1/IKKα, IKK2/IKKβ and NEMO/IKKγ) responsible for the phosphorylation of IκB; such a modification induces the ubiquitination and the proteasomal degradation of IκB, thus rendering NF-κB proteins free to translocate in the nucleus to transactivate responsive genes. The latter is initiated by the activation of NF-κB-inducing kinase (NIK) and IKKα, and the following phosphorylation of the precursor p100, which loses the C-terminal domain and generates the protein p52. Of note, triggering of B cell receptor (BCR), toll-like receptor 4 (TLR4) and receptors for the cytokines TNF-α and IL-1β stimulates canonical NF-κB signaling. Differently, binding to receptors for the cytokines BAFF, RANKL, LTβ and TWEAK activates the noncanonical NF-κB pathway [1].

Oncogenic mutations and inflammation contribute to constitutive activation of NF-κB signaling in many hematologic and solid tumors, including MM [3,4,5].

MM is a neoplasia caused by the expansion and accumulation of terminally differentiated malignant plasma cells (PCs) in the BM. It represents about 10% of the hematologic malignancies with a high incidence in men and in elderly people (median age at diagnosis of 69 years). The main clinical manifestations of MM are indicated by the CRAB criteria: hypercalcemia, renal failure, anemia and bone lesions. In the majority of patients, MM consists in the evolutive phase of a premalignant condition, termed monoclonal gammopathy of undetermined significance (MGUS), with an intermediate asymptomatic stage, named smoldering MM (SMM) [6,7]. The development and progression of MM are strictly dependent on the BM microenvironment in which tumor cells establish mutual interactions with other resident cells or with the extracellular matrix [8]. A relevant cellular component of the BM is characterized by mesenchymal stromal cells (MSCs). They are nonhematopoietic multipotent progenitor cells able to differentiate into adipocytes, chondrocytes and osteocytes. In the BM, healthy MSCs (HD-MSCs) guarantee the development and the maintenance of hematopoietic cells and regulate bone remodeling. The presence of MM significantly alters the functional properties of MSCs which acquire the capability to promote cancer evolution through several mechanisms [9,10]. MSCs from myeloma patients (MM-MSCs) are known to sustain the survival and proliferation of malignant PCs via direct cell-to-cell contacts and the release of many soluble factors (e.g., cytokines, growth factors and extracellular vesicles). Moreover, they favor the induction of osteolytic bone lesions and generation of an immunosuppressive microenvironment [9]. A number of studies demonstrated that NF-κB-dependent inflammatory pathways initiated by TLR4 ligands and cytokines, such as TNF-α and IL-1β, are highly activated in MM-MSCs and are implicated in the changes of these cells in the tumor microenvironment [10,11,12,13]. Moreover, NF-κB is a crucial regulator of the effects that MM-MSCs exert on MM cells, by promoting their growth, chemoresistance and immune escape.

Here, we summarize the latest research on how dysregulation of NF-κB signaling affects MM development and progression. First, we consider the role of these transcription factors in malignant PCs and MM-MSCs autonomous events. Second, we address their contribution in the crosstalk between these cells, focusing on processes responsible for cancer evolution, including MM cell proliferation, drug resistance and recognition by the immune system and MM-associated osteolysis.

## 2. Regulation of MM Cell Activity by NF-κB

### 2.1. Genetic Dysregulation of NF-κB in MM

The NF-κB family of transcription factors represents a major link between cancer and inflammation [14]. In recent years, sequencing of cancer genomes underscored recurrent “gain-of-function” (GOF) mutations in genes encoding for key positive and negative regulators of these proteins. Aberrant activation of NF-κB signaling has been reported in hematological malignancies, and in particular in lymphoid cancers [4,15,16], such as MM. Starting from the past fifteen years, a number of genomic analyses, e.g., from high-array-based comparative genomic hybridization to next-generation sequencing (NGS)-based studies, have investigated NF-κB deregulations in this neoplasia (Figure 1A). NF-κB subunit genes are rarely altered, whereas genes encoding for NF-κB positive and/or negative regulators are often mutated. Indeed, amplification or GOF mutations of MAP3K14 (NIK), TACI, CD40 and LTβ receptors, or inactivation or deletion of TRAF3, CYLD, cIAP1 and cIAP2 have been identified with significant frequencies. Moreover, many mutations affecting other proteins that directly or indirectly control NF-κB functions, such as CARD11, IKBIP, IKBKB, MAP3K1, RIPK4, TNFRSF1A and TLR4, extended the previous findings [17]. Many of these genetic modifications are primarily associated with noncanonical NF-κB signaling. GOF mutations of MAP3K14/NIK, LTβR, CD40, or loss-of-function (LOF) mutations in TRAF3, TRAF2, cIAP1/2 increase the levels of NIK and, consequently, NF-κB2 processing [3]. However, selective IKKβ inhibitors are highly toxic for MM cells, indicating a relevant role mediated by canonical NF-κB activation. These findings highlight relevant variations in the regulatory modalities of classical and alternative NF-κB signaling in selected populations of MM cells.

A further layer of complexity is ascribed to distinct epigenetic landscape changes during MM progression. Alterations in DNA methylation in non-CpG islands characterize the evolution from MGUS to the MM stage. Histone modifications, such as acetylation and methylation, can regulate MM cell differentiation, plasticity and drug response [15,18]. Overexpression of the histone methyltransferase gene EZH2 commonly observed in MM is induced by dysregulated noncanonical NF-κB signaling; here, the inhibition of EZH2 sensitizes MM cells to proteasome inhibitors (PI, e.g., bortezomib), through MYC suppression and inhibition of H3K27 trimethylation, two critical regulators of genes involved in antibody production and B cell metabolism [19]. The identification of mutated NF-κB related genes in MM, and at the different stages of progression, highlights the concept that the activity of this signaling pathway may be heterogeneous across different molecular subtypes of this tumor, with important implications for possible tailored/targeted therapeutic approaches.

### 2.2. NF-κB-Dependent Mechanisms in MM Cell Biology

NF-κB signaling plays a major role in regulating several genes shown to contribute to MM development and progression. High amounts of the NF-κB p65/RelA subunit have been detected in almost 80% of BM biopsies from MM patients, and they correlated with enhanced expression of different antiapoptotic genes regulated by NF-κB. Likewise, p65/RelA, nuclear accumulation of p52 and RelB together with constitutive RelB DNA binding, has been often observed in primary MM samples. Importantly, high NF-κB activity can affect the expression of several cell cycle regulators, such as cyclin E, cyclin D, c-MYC or E2F3α [3,20]. NF-κB signaling can increase cancer cell-intrinsic expression of critical antiapoptotic mediators, including Bcl-xL, Bcl2, cFLIP, cIAP2 and Gadd45β [21] by promoting tumor growth and protecting cancer cells from apoptosis-inducing chemotherapeutic agents. In agreement with these observations, in MM cells with GOF mutations of the noncanonical NF-κB pathway, the activity mediated by RelB and RelB/p50 heterodimers directs the expression of pro-survival and drug-resistance related genes. In a different way, HDAC4-RelB-p52 complexes maintain a repressive chromatin structure around the pro-apoptotic genes *BIM* and *BMF* and regulate MM cell survival and growth. Of note, in this pathway, RelB can be phosphorylated by ERK1 in MM cells, and such a modification is critical for *BIM* gene repression [22]. Both cell-intrinsic and cell-extrinsic processes seem to cooperate in sustaining high NF-κB activity in MM. In this scenario, the open question whether cell-autonomous and activation of NF-κB mediated by microenvironmental factors is strictly integrated in the final action or are mutually exclusive remains unresolved. From a different point of view, NF-κB mutations can arise and be selected in cell clones that become less dependent on extrinsic signals from the BM microenvironment.

## 3. Regulation of MSCs Activity by NF-κB in MM

### 3.1. Mesenchymal Stromal Cells in MM: Properties and Functions

Several studies provided compelling evidence for functional differences between HD-MSCs and MM-MSCs [9,23]. Initial findings highlighted a lower proliferation rate, a reduced osteogenic potential, and an altered production of many soluble factors by MM-MSCs compared to healthy counterparts. Later, significant abnormalities emerged by gene expression analysis of both cultured expanded in vitro or freshly isolated MM-MSCs [10,11,24,25,26,27]. Remarkably, MSCs are characterized by a specific signature consisting in a differential expression of genes encoding for proteins involved in the regulation of tumor growth, angiogenesis and osteoblast/osteoclast (OBs/OCs) differentiation, but also associated with an inflammatory signaling. Gene expression patterns of MM-MSCs vary among disease stages and clinical conditions (type of organ dysfunctionality (e.g., lytic bone lesions or renal failure) and relapsed and/or refractory MM). These findings provide evidence of the strong impact of MSCs on the evolution of this neoplasia and suggest that MSCs represent an independent prognostic factor for clinical outcome

### 3.2. Inflammatory MM-MSCs and NF-κB Signaling

Inflammation is a hallmark of tumor microenvironment including the MM BM. Several lines of evidence described an MM-associated inflammatory network that also significantly affects and involves MSCs. Among complex mechanisms of inflammation, a major role is played by the cytokines and the pattern recognition receptors (PRRs). Studies of transcriptomic analysis demonstrated the existence of myeloma-specific inflammatory MSCs (iMSCs) characterized by an enriched expression in TNF-α and IL-1β signaling [10,11]. Moreover, among PPRs, TLR4 pathway activation emerged as a crucial regulator of iMSCs in MM [12,13]. Of note, NF-κB is the main downstream effector of these inflammatory pathways, thus indicating the capability of these transcription factors to regulate iMSCs activity in MM (Figure 1B).

By using a single-cell RNA sequencing approach, de Jong and colleagues have recently identified two clusters of MSCs in MM patients but not in healthy donors [11]. Based on the enrichment for the pathway “TNF-α signaling via NF-κB”, consisting in increased transcription of genes encoding for cytokines (IL-6 and LIF) and chemokines (CXCL2, CXCL3, CXCL5, CXCL8)*,* these cells were defined as iMSCs. Interestingly, iMSCs transcriptome persists in treated patients either negative or positive for minimal residual disease. This observation indicates that antitumor therapy does not revert BM inflammation, suggesting that iMSCs guarantee an inflammatory context which promotes the relapse of disease. It was also hypothesized that TNF-α and IL-1β, produced by T or natural killer (NK) cells and myeloid cells, respectively, may be responsible for the activation of iMSCs. Indeed, they express high levels of genes encoding for the receptor of these cytokines (TNFRSF1A, TNFRSF1 and IL1R), and HD-MSCs can acquire an inflammatory phenotype upon ex vivo treatment with TNF-α and IL-1β. Consistently, a study of spatial distribution revealed the interactions of iMSCs with both malignant PCs and T lymphocytes [11]. These findings demonstrated that MSCs are active players of inflammation in MM BM by supporting tumor survival and modulating the immune response.

MSCs constitutively express TLR1–6 and the engagement of these receptors can affect their activity [28]. Growing evidence showed that TLR4 expression is altered in both malignant PCs and MM-MSCs [12,28]. TLR4 expression is higher in MM-MSCs than HD-MSCs with significant differences among disease stages; indeed, the levels of this PPR seem to be reduced in MSCs from patients with complete remission but they are augmented in case of relapse. In addition, the expression of the adaptor molecule MyD88, which is necessary for TLR4 signaling, is increased in MM-MSCs as compared with HD-MSCs [12]. These findings suggested that TLR4 expression and signaling in MM-MSCs parallels the clinical condition. Concerning the function of this receptor, in vitro stimulation of MM-MSCs with TLR4 ligands increases the expression of the molecules required for the interaction with MM cells and for their survival, growth and chemoresistance, such as the adhesion molecules CD49e and CD54 and the cytokine IL-6 [13]. Consistently, a selective inhibitor of TLR4 signaling, C34, limits the capability of MSCs to sustain MM cell proliferation showing also additive effects with the antimyeloma drug lenalidomide. As well, C34 administration significantly increases the survival rate in a Vκ*MYC mouse model of MM, indicating that specific targeting of tumor microenvironment can compromise the development of this neoplasia. These observations confirmed data obtained in a model of MM in the adult zebrafish, where TAK-242, a compound disrupting TLR4 signaling, decreases malignant PCs engraftment [12].

As mentioned above, different stages of disease have been associated with alterations in gene expression of MM-MSCs, but it remains unclear if properties of iMSCs can also change and if they are already present in MGUS condition. Understanding these aspects could be helpful to better define the contribution of these cells in the development and/or progression of MM and to identify possible therapeutic targets and prognostic factors. 

## 4. Regulation of the Crosstalk between MM Cells and MSCs by NF-κB Pathway

### 4.1. MSCs-MM Crosstalk and Tumor Proliferation: Role of NF-κB

The role of MSCs in supporting MM cell survival and growth is well recognized (Figure 2). NF-κB contributes to these processes via direct regulation of genes controlling cell cycle and apoptosis as well as of genes encoding for adhesion molecules, cytokines and growth factors able to additionally increase MM cell proliferation. Direct adhesion to MSCs represents a strong stimulus for NF-κB pathway activation in MM cells. These malignant cells express on their surface the lymphocytes function-associated antigene-1 (LFA-1) and very late antigen (VLA-4) that bind the intercellular adhesion molecule 1 (ICAM-1) and vascular adhesion molecule 1 (VCAM-1) on MSCs. These adhesive contacts trigger the NF-κB pathway in MSCs resulting in the transcription and secretion of a variety of cytokines including IL-6, a major growth factor for MM cells [29]. Moreover, in the BM milieu, several cytokines can act as autocrine or paracrine prosurvival factors for MM cells. An example is given by TNF-family ligands, a proliferation-inducing ligand (APRIL) and B cell activating factor (BAFF), which are abundantly present in MM-MSCs and osteoclasts. Triggering of these molecules with their cognate BAFFR and TACI/BCMA receptors facilitates cell adhesion and survival of MM cells through NF-κB signaling and upregulation of the antiapoptotic factors MCL-1 and BCL-2 [30]. Accordingly, blockade of BAFF and APRIL activity via atacicept (TACI-Ig, a fusion protein containing the ligand-binding domain of the receptor TACI and the Fc portion of hIgG1) and BCMA decoy receptor (sBCMA-Fc), can be sufficient to reduce tumor burden in mouse models of MM [31,32]. It was also demonstrated that growth-arrest-specific gene 6 (GAS6), a cytokine overproduced by both MM-MSCs and MM cells, is required for the proliferation and survival of MM cells [33]. Like protein S, GAS6 binds the receptors TYRO3, AXL and MERTK (TAM receptors), which induce PI3K/AKT, ERK1/2, NF-κB and p38 signaling [34]. *GAS6* silencing in MM cells results in reduced proliferation and increased cell death, and consistent knockdown of *MERTK* prolongs the survival of myeloma-bearing mice [35]. Moreover, a GAS6-neutralizing antibody prevents the ability of MSCs-derived soluble factors to sustain MM survival by suppressing the NF-κB signaling pathway and IL-6 expression [36]. It is likely that most of these cytokines also account for the proliferative effects of MSC-derived extracellular vesicles (EVs), such as exosomes and microvesicles (MVs). Indeed, EVs are important carriers of proteins, lipids and nucleic acids and mediate the crosstalk between BM microenvironment and malignant PCs, supporting angiogenesis, osteolysis and drug resistance [37]. MM-derived exosomes induce NF-κB activation in MSCs with increased secretion of IL-6 [38], while MVs released from MM-MSCs, but not HD-MSCs, stimulate the translation initiation and increase the survival and proliferation of MM cells. These effects are associated with high levels of proteins with established roles in MM pathogenesis, including NF-κB, SMAD5, cyclin D, HIF1α and cMYC [39].

### 4.2. MSCs-MM Crosstalk and Bone Disease: Role of NF-κB

One of the clinical complications of MM is osteolysis, a process which allows tumor cells to accumulate within the BM. A general understanding is that the balance between osteoclastogenesis and osteoblastogenesis is altered in MM patients [40]. Indeed, osteolytic lesions are due to the increased bone resorption paralleled to reduced bone formation. At the molecular level, key mediators of these processes include receptor activator of NF-κB ligand (RANKL), osteoprotegerin (OPG) system (RANKL/OPG), wingless (Wnt) and Dickkopf-1 (Wnt/DKK1) pathway. RANK, its ligand RANKL, and the decoy receptor OPG are major positive regulators of OCs formation from their precursors together with their activity. Wnt proteins promote OBs differentiation, whereas DKK1 inhibits osteoblastogenesis.

A dysregulated BM microenvironment has a critical role in MM bone disease. Here, we focus on the contribution of MSCs-MM crosstalk in these mechanisms with a particular attention for NF-κB-mediated events (Figure 3).

NF-kB proteins are key regulators of osteoclastogenesis [41]. These transcription factors are mainly activated downstream from RANK-RANKL interaction, which is determinant for OCs differentiation from precursor cells. In MM, enhanced osteoclastogenesis is mainly promoted by MSCs ability to produce high levels of RANKL and IL-6. The expression of these cytokines is regulated by TRAF6, an E3 ubiquitin ligase responsible for the K63-linked polyubiquitination and activation of the serine/threonine kinase TGF-β-activated kinase-1 (TAK1). Indeed, TRAF6-dominant negative peptides and TAK1 inhibitors (TAK1i, e.g., LLZ1640) are sufficient to decrease tumor burden while preventing dysregulated bone resorption in vivo [42,43]. In this scenario, TAK1 represents an attractive therapeutic target to reduce at various levels the consequence of disrupted bone homeostasis in MM. In the BM, MM cells release other factors that cooperate with RANK/RANKL to promote OCs differentiation and activity. In particular, the secretion of hepatocyte growth factor (HGF) by MM cells can regulate bone remodeling by promoting OCs differentiation and limiting OBs activity. HGF secreted by MM cells induces RANKL production by murine MSC and OB cell lines through the activation of the Met/NF-κB signaling pathway [44]. Interestingly, the c-Met inhibitor ARQ-197 limits MM cell proliferation in vitro and in a JJN3-NSG xenograft model, and this parallels with a reduced number of osteolytic lesions [45]. Generation of bone lesions are also promoted by the antiosteoblastogenic activity of MM cells. They inhibit OBs maturation from precursors, by secreting DKK1, frizzled-related protein 2 (sFRP-2) and sclerostin; these factors block the Wnt pathway activation, which is responsible for runt-related transcription factor 2 (RUNX2)-induced OBs differentiation [46]. A further layer of complexity in the regulation of MM-induced bone disease can be provided through the release of EVs, which affects OBs and/or OCs activity. Through the activation of the Ape1/NF-κB pathway, MM cell-derived exosomes elicit IL-6 production and inhibit Runx2, osterix and osteocalcin expression in MSCs, thus avoiding their differentiation toward OBs [38]. Moreover, IL-32^+^ EVs released by MM cells in response to hypoxia drive OCs differentiation via induction of NF-κB-mediated expression of pro-osteoclastogenic transcription factors [47]. Consistently, MM cell-derived exosomes induce osteolysis in vivo and GW4869, an inhibitor of extracellular vesicles secretion, synergizes with bortezomib in reducing tumor load and decreasing bone destruction in C57BL6/KalwRij mice [48].

### 4.3. MSCs-MM Crosstalk and Anti-MM Immune Response: Role of NF-κB

BM microenvironment is an important reservoir of innate and adaptive immune cells that are fundamental to control MM at the early stage of disease. However, immune effector cells undergo several phenotypic and functional changes rendering them unable to prevent disease progression [49,50,51]. As well, malignant PCs develop various mechanisms to escape from antitumor immune responses [52]. The interactions with other cells in the BM microenvironment are crucial to avoid immune-mediated recognition of MM cells and to inhibit the activity of immune cells, with MSCs playing a major role [11,53]. A mechanism consists in the regulation of the expression of immune activating and inhibitory ligands on MM cells affecting their susceptibility to effector lymphocytes (Figure 4).

Our group demonstrated that MSCs-induced NF-κB pathway activation is required for the increased expression of NK cell-activating ligands in MM cells [54,55]. MSCs induce significant upregulation of MHC class I chain-related protein A (MICA) and poliovirus receptor (PVR) levels on these cancer cells. Consequently, they are better recognized by NK cells via engagement of the activating receptors natural-killer group 2 member D (NKG2D) and DNAX accessory molecule-1 (DNAM-1), respectively. The underlying mechanism is the enhanced activity of NF-κB transcription factors in response to GAS6 and IL-8-bearing MVs released by MSCs [56,57]. Although activating ligands, MICA and PVR can also inhibit NK cell activity. MM progression is associated with reduced surface levels of MICA due to proteolytic cleavage [58]. Soluble isoform of MICA (sMICA) and MICA-expressing EVs induce internalization of surface NKG2D and cause an impairment of NK cell effector functions [59]. Concerning PVR, as an adhesion molecule, it may contribute to support MM cell survival and proliferation by enhancing the direct contact with MSCs. Furthermore, PVR can also bind the inhibitory receptors CD96 and TIGIT, which are enhanced on NK cells during MM progression [48]. Consistently, PVR overexpression is correlated with tumor progression and unfavorable prognosis in different cancer cell types [60,61]. It is likely that the effect on antitumor immune response of MSCs-mediated MICA and PVR upregulation on malignant PCs varies in distinct disease stages. Increased expression of these ligands may render MM cells more susceptible to NK cell-mediated attack at early stage of disease, while leading to a defective activity of these effector lymphocytes during disease progression. 

### 4.4. MSCs-MM Crosstalk and Drug-Resistance: Role of NF-κB

Advances in therapeutic approaches have improved MM patient outcomes [6]; however, this neoplasia remains incurable. Failure of treatments is mainly caused by the development of multidrug resistance (MDR) [62,63], which is largely dependent on the interactions of malignant PCs with the BM microenvironment, especially with MSCs. Cancer cells are protected by the cytotoxic effects of alkylating agents (melphalan), PI and immunomodulatory drugs (IMiDs, lenalidomide and pomalidomide) by soluble and cell–cell mediated signals derived from MSCs [64]. Thus, targeting these molecular pathways offers a potential therapeutic strategy to overcome drug resistance.

NF-κB activity is critical for the intrinsic or MSCs-induced chemoresistance in MM (Figure 5). Constitutive NF-κB pathway activation accounts for MM cell unresponsiveness to several classical drugs, such as melphalan and dexamethasone [65,66,67] and to novel pharmacologic agents under current investigation, such as aurora kinase inhibitors [68]. Furthermore, direct adhesive contacts with MSCs and a number of cytokines (e.g., TNF-α, IL-1β, IL-6, IL-8 and BAFF) released by these cells contribute to induce NF-κB activation and to render MM cells resistant both to PI and/or IMiDs, suggesting the possibility to combine these drugs with NF-κB inhibitors and/or antibodies blocking these soluble factors [69]. These effects have been intensively reviewed [3,4] and below we focus on the latest reports.

A recent study highlighted the molecular mechanisms underlying IL-6 and DKK1-induced bortezomib-resistance [70]. IL-6 stimulates MM cells to express CKAP4, the receptor for DKK1. The interaction DKK1/CKAP4 then evokes the activation of noncanonical NF-κB signaling and protects tumor cells from bortezomib-induced apoptosis. Consistently, serum levels of DKK1 and CKAP4 expression are higher in refractory/relapsed MM patients than in those with a complete clinical response. In addition, anti-DKK1 and IL-6 neutralizing antibodies improve sensitivity to bortezomib in a mouse model of MM. IL-8, TNF-α and IL-1β can be involved in these mechanisms both as direct activators of NF-κB and/or regulators of IL-6 expression in MM cells. A role in the induction of bortezomib-resistance of MM cells has been also proposed for the hyaluronan and proteoglycan link protein 1 (HAPLN1) [71], a protein of extracellular matrix. MM-MSCs express high levels of HAPLN1, generally cleaved by the matrix metalloproteinase 2 to release a small fragment, which is able to enhance NF-κB activation in MM cells, by reducing PI-induced apoptotic cell death. Among MSCs-derived soluble factors, a major role in the induction of IMiDs resistance is played by TNF-α [72]. The cytokine is responsible for the proteasomal degradation of TRAF2, which upregulates NIK expression and triggers the alternative NF-κB pathway activation. These transcription factors contribute to IMiDs resistance mainly through induction of ERK signaling. Accordingly, inhibition of MEK-ERK by AZD6244 can overcome IMiDs resistance in vitro and in an inducible TRAF2 knockdown MM xenograft model. In support of this evidence, inactivating mutations of TRAF2 in MM patients cause the constitutive activation of the noncanonical NF-κB pathway, and TRAF2 expression is lower in IMiDs refractory-MM patients than in newly diagnosed patients still responsive to these drugs. These observations provide the rationale for using inhibitors of NF-κB and/or MEK/ERK signaling to overcome IMiDs resistance and improve MM patient outcome in MM.

MDR in MM is a very complex phenomenon and many aspects remain to be elucidated. A multitude of molecules concur to generate a resistant phenotype to different chemotherapies in malignant PCs. Remarkably, most of them are also important mediators of other pathologic aspects of this neoplasia and NF-κB is often a key player of these mechanisms. On one hand, these data suggest the necessity of an anti-MM therapeutic approach based on concomitant targeting of different pathways, and on the other hand they offer the possibility to focus treatments on a common signaling represented by NF-κB.

## 5. Concluding Remarks

Our knowledge of MM biology has significantly improved over the past decades. It is increasingly evident that the BM microenvironment plays a crucial role in the development and progression of this neoplasia. MSCs exert a strong supportive function for the expansion of malignant PCs. Among complex signaling pathways involved in these mechanisms, the NF-κB family of transcription factors is a key driver of cell autonomous changes in both MSCs and MM cells as well as of the reciprocal interactions between these cells (Table 1). Therefore, blockade of the NF-κB pathway could be a promising therapeutic target for MM. To date, several inhibitors have been proposed to suppress this pathway, such as peptides, small molecules, siRNAs and antibodies [73,74]. Recent studies describing novel compounds able to block this pathway in MM are listed in Table 2. A great limit in using these approaches is represented by their toxic side-effects. Indeed, NF-κB controls many physiological processes, including the development and the activity of many immune cells [75]. This aspect has critical relevance in a tumor context where boosting of immune response would be desirable. For MM, the complexity of the mechanisms underlying NF-κB activity in malignant PCs together with the high heterogeneity of this neoplasia renders difficult the blockage of this pathway. An approach to selectively target tumor cells based on concomitant inhibition of canonical and noncanonical NF-κB signaling may be helpful. Moreover, findings discussed in this review suggest an extension of this approach to MSCs. To this aim, the introduction of nanocarriers used as vehicles for the delivery of NF-κB inhibitors and antitumor agents [73] to improve the bioavailability of drugs and selective targeting of these cells is very promising and encouraging.

## Figures and Tables

**Figure 1 ijms-24-01823-f001:**
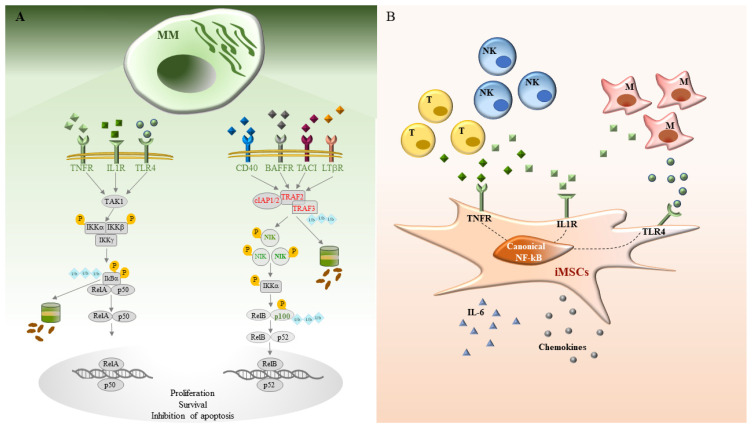
(**A**) Regulation of MM cell activity by NF-κB. Schematic representation of classical (on the left) and alternative (on the right) activation pathways of NF-κB signaling. Gain of functions (green) and loss of functions (red) mutations of the NF-κB pathway are indicated. (**B**) Regulation of MSCs activity by NF-κB. NF-κB-dependent inflammatory pathways initiated by TLR4 ligands and cytokines, such as TNF-α and IL-1β, characterize myeloma specific inflammatory MSCs (iMSC).

**Figure 2 ijms-24-01823-f002:**
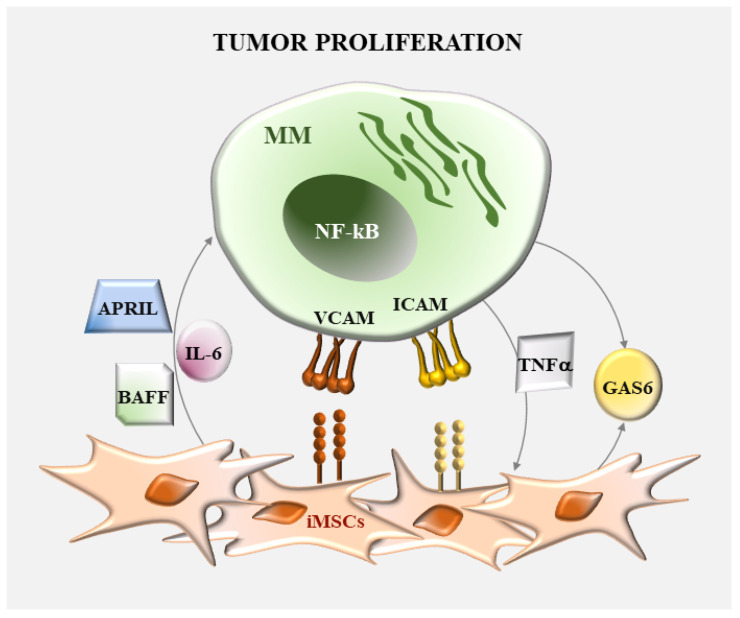
Role of NF-κB in the regulation of MM cell proliferation by MSCs. The survival and proliferation of malignant plasma cells in the bone marrow microenvironment is sustained by adhesive contacts and soluble factors produced by MSCs.

**Figure 3 ijms-24-01823-f003:**
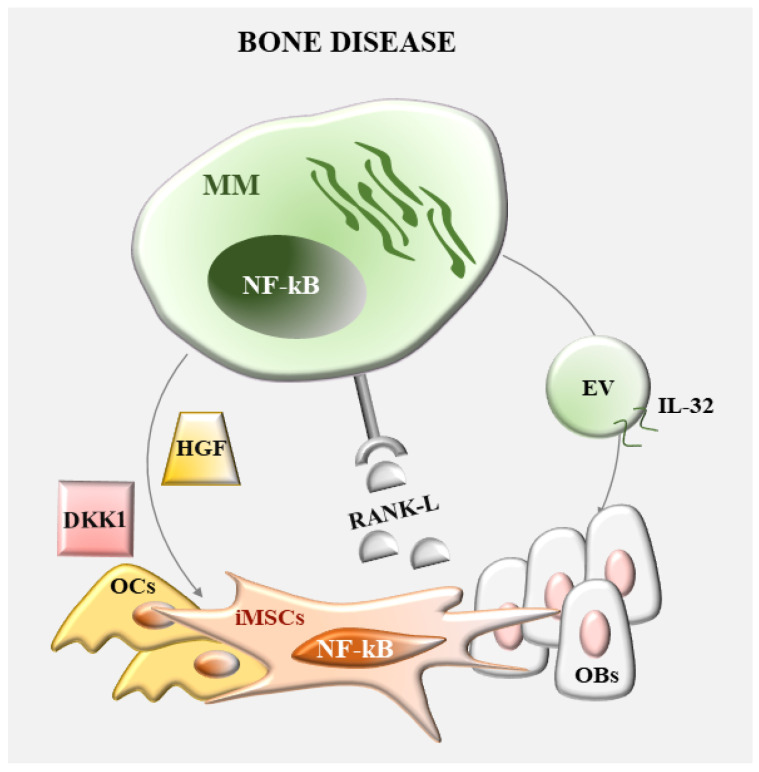
Role of NF-κB in the regulation of MM bone disease by MSCs. Main NF-κB dependent mechanisms responsible for increased osteoclastastogenesis and reduced osteoblastogenesis in MM. Osteoclasts (OCs); osteoblasts (OBs).

**Figure 4 ijms-24-01823-f004:**
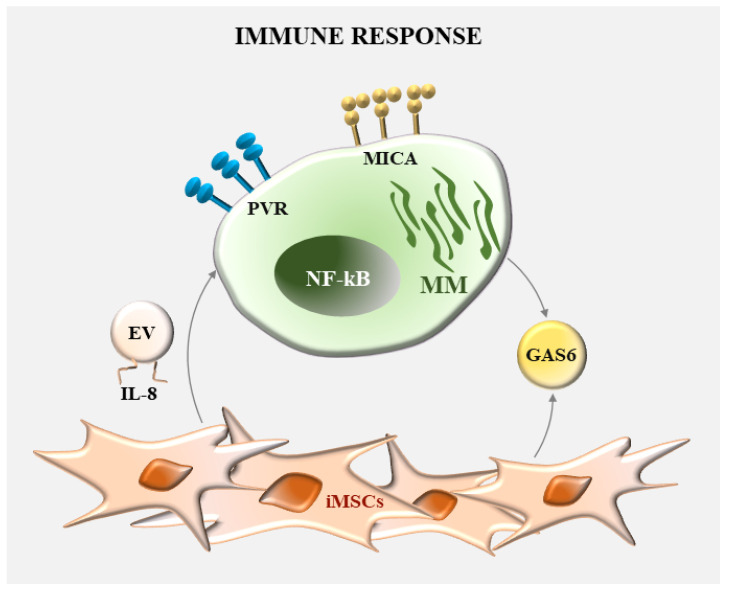
Role of NF-κB in the regulation of antimyeloma immune response by MSCs. Expression of immune-activating ligands MICA and PVR is regulated by cytokines released by MSCs.

**Figure 5 ijms-24-01823-f005:**
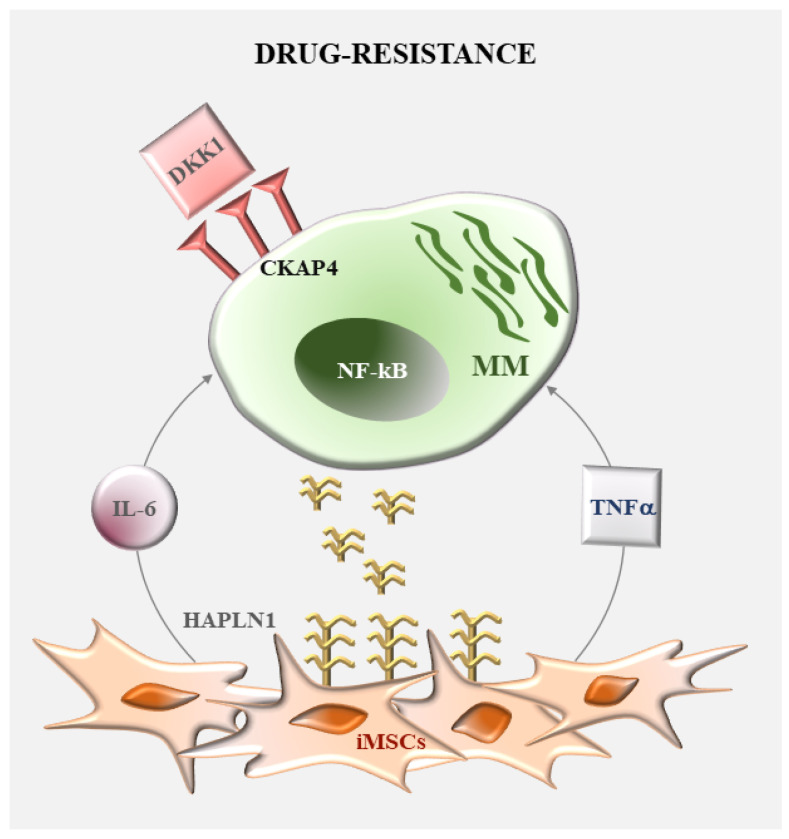
Role of NF-κB in MM drug resistance. MSC-derived soluble factors involved in the induction of MM resistance to bortezomib (in gray) and IMiDs (in blue).

**Table 1 ijms-24-01823-t001:** NF-kB-dependent mechanisms in the crosstalk between multiple myeloma (MM) and mesenchymal stromal cell (MSCs). Osteoblasts (OBs); osteoclasts (OCs); extracellular vesicles (EV); microvesicles (MVs).

	Activating Stimulus	Functional Effect	Reference
**MM-MSCs CROSSTALK**	IL-6 and DKK1 HAPLN1	Bortezomib resistance	[70,71]
	TNFα	IMiDs resistance	[72]
	GAS6	MICA upregulation	[56]
	IL-8^+^ MVs	PVR upregulation	[57]
	ICAM-1, VCAM-1 GAS6 Exosomes MVs	IL-6 production and MM proliferation	[37,39]
	APRIL, BAFF	MCL-1 and BCL-2 upregulation	[30]
	RANKL	Increased OCs differentiation	[43]
	HGF	Increased OCs differentiation	[44,45]
	DKK1, sFRP-2 and Sclerostin	Reduced OBs maturation	[46]
	Exosomes	Reduced OBs differentiation and bone mineralization Apoptosis of OBs precursors Increased OCs resorptive activity	[38]
	IL-32^+^ EVs	Increased OCs differentiation	[47]

**Table 2 ijms-24-01823-t002:** Novel compounds targeting the NF-kB signaling pathway in MM cells.

Compound	Molecular Target	Type of Study	Trial N	Reference
Indirubin-3′-monoxime	Proteasome	In vitro	-	[76]
Isoginkgetin	Proteasome	In vitro	-	[77]
TL32711	cIAP	In vitro In vivo (mouse model)	-	[78]
LCL161	cIAP	In vitro In vivo (mouse model)	NCT01955434 NCT03111992	[79]
LLZ1640-2	TAK1	In vitro In vivo (mouse model)	-	[43]
MLN4924	NEDD8-activating enzyme	In vitro	NCT03770260 NCT00722488	[80]
TAS-4464	NEDD8-activating enzyme	In vitro In vivo (mouse model)	NCT02978235	[81]
TAK-242	TLR4	In vitro In vivo (zebrafish model)	-	[12,82]
C34	TLR4	In vitro In vivo (mouse model)	-	[13]
sBCMA-Fc	APRIL and BAFF	In vitro In vivo (mouse model)	-	[32]
ARQ-197	c-Met	In vitro In vivo (mouse model)	NCT01447914	[45]
IT848	NF-kB	In vitro In vivo (mouse model)	-	[83]
DTP3	GADD45/MKK7	In vitro 3 MM patients	-	[84,85]

## Data Availability

Not applicable.

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
