# Peer review of "Role of NF-κB Signaling in the Interplay between Multiple Myeloma and Mesenchymal Stromal Cells"

_ijms, 2023, doi:10.3390/ijms24031823_

Round 1

Reviewer 1 Report

The Authors have done good job by writing a review manuscript on the role of Nuclear Factor-κB (NF-κB) transcription in multiple myeloma (MM). 

However, the manuscript needs some changes to make it better and comprehensive.

1. To check the scientific and English language.

2. Adding some information on current treatments and role of targeting the NF-κB pathways and current drugs available and in clinical trials in the tabular form or Figure.

Author Response

We thank the Reviewers for the suggestions which improved our manuscript. A point-to-point response is provided below.

Reviewer 1

The Authors have done good job by writing a review manuscript on the role of Nuclear Factor-κB (NF-κB) transcription in multiple myeloma (MM). 

However, the manuscript needs some changes to make it better and comprehensive.

1.To check the scientific and English language.

-To address the reviewer’s point regarding the language, we changed unclear or syntactically incorrect sentences.

  1. Adding some information on current treatments and role of targeting the NF-κB pathways and current drugs available and in clinical trials in the tabular form or Figure.

-As suggested by the Reviewer, we listed novel compounds shown to target (directly or indirectly) NF-kB signaling pathway in MM cells in the Table 2.

Reviewer 2

Thank you for inviting me to review the manuscript by Marco Cippitelli et al., that covers an interesting topic but, in my opinion, requires a major revision:

  1. Please do not use abbreviations as keywords. Keywords should facilitate the association with MeSH terms.

- As requested by the Reviewer, we enclosed only MeSH terms as keywords.

  1. Spelling and grammar errors are still included in the manuscript.

- We apologize for these mistakes. We corrected spelling and grammar errors.

  1. Sections “Introduction” and “Discussion” should be written based on the very recently papers published on that subject. Please revise the cited papers and include current papers from the last 4 years.

- We follow the Reviewer’s suggestion and revised citations including mainly papers from the last 5 years.

  1. Subsections (4.1 – 4.4) in “Regulation of the crosstalk between MM and MSCs by NF-κB pathway” section are too long and should be more systematized and divided. Moreover, each subsection should have a separate figure.

- As suggested by Reviewer, we edited Section 4 and associated a single Figure to each subsection.

Reviewer 2 Report

Dear Editor,

Dear Authors,

Thank you for inviting me to review the manuscript by Marco Cippitelli et al., that covers an interesting topic but, in my opinion, requires a major revision:

1. Please do not use abbreviations as keywords. Keywords should facilitate the association with MeSH terms.

2. Spelling and grammar errors are still included in the manuscript.

3. Sections “Introduction” and “Discussion” should be written based on the very recently papers published on that subject. Please revise the cited papers and include current papers from the last 4 years.

4. Subsections (4.1 – 4.4) in “Regulation of the crosstalk between MM and MSCs by NF-κB pathway” section are too long and should be more systematized and divided. Moreover, each subsection should have a separate figure.

Author Response

(The authors gave the same response as above.)

Round 2

Reviewer 2 Report

The manuscript by M. Cippitelli et al. has been revised according to the comments and in my opinion can be published in its current version.